# The Rapid Evolution of Resistance to Vip3Aa Insecticidal Protein in *Mythimna separata* (Walker) Is Not Related to Altered Binding to Midgut Receptors

**DOI:** 10.3390/toxins13050364

**Published:** 2021-05-20

**Authors:** Yudong Quan, Jing Yang, Yueqin Wang, Patricia Hernández-Martínez, Juan Ferré, Kanglai He

**Affiliations:** 1Instituto de Biotecnología y Biomedicina (BIOTECMED), Department of Genetics, Universitat de València, 46100 Burjassot, Spain; quanyu@alumni.uv.es (Y.Q.); patricia.hernandez@uv.es (P.H.-M.); 2State Key Laboratory for Biology of Plant Diseases and Insect Pests, Institute of Plant Protection, Chinese Academy of Agricultural Sciences, No. 2, West Yuanmingyuan Road, Beijing 100193, China; yangjing@biols.ac.cn (J.Y.); wangyueqin@caas.cn (Y.W.)

**Keywords:** *Bacillus thuringiensis*, Bt toxins, cross resistance, oriental armyworm

## Abstract

Laboratory selection for resistance of field populations is a well-known and useful tool to understand the potential of insect populations to evolve resistance to insecticides. It provides us with estimates of the frequency of resistance alleles and allows us to study the mechanisms by which insects developed resistance to shed light on the mode of action and optimize resistance management strategies. Here, a field population of *Mythimna separata* was subjected to laboratory selection with either Vip3Aa, Cry1Ab, or Cry1F insecticidal proteins from *Bacillus thuringiensis*. The population rapidly evolved resistance to Vip3Aa reaching, after eight generations, a level of >3061-fold resistance, compared with the unselected insects. In contrast, the same population did not respond to selection with Cry1Ab or Cry1F. The Vip3Aa resistant population did not show cross resistance to either Cry1Ab or Cry1F. Radiolabeled Vip3Aa was tested for binding to brush border membrane vesicles from larvae from the susceptible and resistant insects. The results did not show any qualitative or quantitative difference between both insect samples. Our data, along with previous results obtained with other Vip3Aa-resistant populations from other insect species, suggest that altered binding to midgut membrane receptors is not the main mechanism of resistance to Vip3Aa.

## 1. Introduction

Oriental armyworm, *Mythimna separata* (Walker) (Lepidoptera: Noctuidae) (Appendix A) is a polyphagous pest that can feed on more than 300 species, including some important ones, as a staple food, such as corn, rice, and wheat [1,2]. It is a major migratory agricultural insect pest in East Asia. Outbreaks of *M. separata* have caused devastating damage to grain production and economic losses [3]. For control of this pest, various chemical pesticides, mainly organophosphates and pyrethroids, have been heavily employed, which consequently lead to resistance in field populations, as was reported in Shaanxi and Shanxi Provinces of China [4]. *Bacillus thuringiensis* (Bt) insecticidal proteins are a good alternative to synthetic insecticides and their use is being considered in the suppression of this pest [5]. Bt-based insecticides account for 75–95% of the microbial biopesticide market, and Bt crops (genetically modified crops expressing Bt insecticidal proteins) are being commercialized globally [6]. However, as with any other insecticide, field populations of target pests have already developed resistance to Bt insecticides and Bt crops due to their extensive and long-term use [7,8,9].

Thus far, all cases of reported field resistance to Bt insecticides or Bt crops were due to resistance to the Cry proteins [7,8,9]. Vip3 proteins are a different class of Bt insecticidal proteins produced during the vegetative growth phase, which do not share sequence homology and binding sites with Cry proteins [10]. Vip3A proteins have high activity and specificity against lepidopteran pests, and Vip3Aa19 and Vip3Aa20 have been first expressed as single insecticidal proteins in cotton and corn [10,11]. More recently, Vip3A proteins are being used in the pyramided strategy of combining them with Cry proteins in the same crop to increase activity and delay insect resistance [12]. Although practical field resistance to Vip3 proteins has never been reported, a relatively high frequency of Vip3Aa resistant alleles has been detected in Australian populations of *Helicoverpa armigera* and *Helicoverpa punctigera* [13] and in US populations of *Helicoverpa zea* [14]. Laboratory selection has also rendered Vip3Aa resistance in other insect species such as *Heliothis virescens* [15], *Spodoptera litura* [16], and *Spodoptera frugiperda* [17]. However, no mechanistic data on field- or laboratory-evolved resistance to Vip3 proteins have ever been reported. Understanding the genetic and biochemical basis of resistance is crucial in the development of strategies to delay or prevent the evolution of resistance.

In this study, we aimed to determine the potential of an important Asian agricultural pest species, *M. separata*, to evolve resistance against three Bt proteins expressed in Bt crops: Vip3Aa, Cry1Ab, and Cry1Fa. Since resistance was rapidly developed only against Vip3Aa, cross resistance against the Cry proteins and Vip3Aa binding to brush border membrane vesicles (BBMV) from larvae midguts were analyzed.

## 2. Results

### 2.1. Response to Selection with Vip3Aa and Cry1 Toxins in M. separata

Bioassays with Vip3Aa, Cry1Ab, and Cry1F proteins with a recently established *M. separata* population rendered LC_50_ values of 0.86, 0.14, and 1.45 µg/cm^2^ for Vip3Aa, Cry1Ab, and Cry1F, respectively (Appendix A). Three different lines were selected with Vip3Aa, Cry1Ab, and Cry1F separately. Immediately upon start of the selection regimes, the sensitivity of *M. separata* to Vip3Aa decreased significantly (Appendix A). As early as in one or two generations of selection, the LC_50_ value of Vip3Aa was 26.8 and 238 µg/cm^2^, which was about 31- and 278-fold higher, compared with the susceptible control insects. At the eighth generation of continuous selection, the LC_50_ value was >1684 µg/cm^2^, with an at least 3061-fold resistance ratio (RR_50_) relative to the unselected population (Table 1). On the contrary, in the other two selection lines, no significant change in the susceptibility to the Cry1 proteins (Cry1Ab or Cry1F) was observed during the eight or ten generations of selection (Table 1, Appendix A). Considering that the laboratory population started with 2500 eggs collected from the field, and at the very least one allele for Vip3Aa resistance had to be present in that sample, we can make an estimate of the frequency of Vip3Aa resistance alleles of 0.0002 (1 in 5000 gene copies). However, because of the rapid increase in resistance in the very early generations of selection, the initial allele frequency must have been higher than this value.

### 2.2. Cross-Resistance Evaluation of Vip3Aa-Resistant M. separata Strain to Cry1 Proteins

To determine whether the subpopulation that responded to the Vip3Aa selection could have also developed cross resistance to the Cry1 toxins, the Vip3Aa selected insects were tested with Cry1Ab and Cry1F in the ninth generation. The results showed that LC_50_ of Cry1Ab toxin was significantly increased to 0.39 µg/cm^2^ in the ninth Vip3Aa-selected strain (Ms-R) (95% FL overlap test). However, we did not detect any difference of Cry1F in the Ms-S and Ms-R strain, and the LC_50_ value was about 1.15 µg/cm^2^ (Table 2).

### 2.3. ^125^I-Vip3Aa Binding to BBMV of Susceptible and Resistant M. separata

Specific binding of Vip3Aa to *M. separata* BBMV was shown by incubating a fixed amount of labeled ^125^I-Vip3Aa with increasing concentrations of BBMV, in the presence or absence of an excess of unlabeled Vip3Aa (Figure 1a). Around 20% of ^125^I-Vip3Aa (total binding) used in the assay bound to the BBMV, of which approximately 70% was specific. BBMV from the susceptible and resistant insects showed similar profiles and percent of specific binding, indicating that the resistance of Vip3Aa was not due to an absence of binding to the epithelial membrane. This result was further confirmed by performing competition binding assays (Figure 1b). Again, very similar curves were obtained giving estimated equilibrium dissociation constants (*K*_d_) and concentration of binding sites (*R*_t_) very similar and non-significantly different (Table 3).

## 3. Discussion

Monitoring resistance is important for the widespread and long-term use of commercial adoption of Bt crops since it provides information on how likely is that insect pest populations can evolve resistance to the insecticidal proteins. Resistance to Vip3Aa has been reported in several species [13,14,15,16,17], in all cases, after laboratory selection of field populations. In the present study, we subjected three subpopulations of *M. separata* to laboratory selection with Vip3Aa, Cry1Ab, or Cry1F proteins, all of them already expressed in Bt corn and highly effective against corn pests [12,18,19,20,21]. Only Vip3Aa showed a rapid response to selection, whereas the Cry1 proteins were unable to elicit such a response. A similar rapid response to Vip3Aa selection was also found in *H. virescens*, reaching a >2300-fold level of resistance at the tenth generation [15]. This rapid evolution of selection under laboratory conditions is in contrast with results obtained with Cry1 proteins, such as in this work and others. A Cry1Ab-resistant population of *Ostrinia furnacalis* (Guenée) acquired around 100-fold resistance level only after 35 generations of selection [22] and an *Ostrinia nubilalis* population developed more than 3000-fold resistance to Cry1F after 35 generations of selection [23]. This difference in response to selection, in addition to reflecting a much higher frequency of resistance alleles for Vip3Aa, may suggest differences in the mechanisms of resistance to Vip3Aa and Cry1 proteins. Our results also detected no significant cross resistance to the Cry1 proteins in the Vip3Aa-selected line of *M. separata*.

Many studies have reported that the alteration of membrane receptors is a common evolutionary mechanism conferring high levels of resistance to Cry proteins, such as mutations in the aminopeptidases N, cadherin, or ABC transporters that serve as putative receptors for Cry proteins [7,8]. However, Vip3 proteins do not share binding sites with Cry proteins [10,24,25,26,27]. Although some proteins (from Sf9 or Sf21 cells) have been identified to bind the Vip3Aa protein and thus to serve as receptors [28,29,30], their relationship with Vip3A resistance has never been established. Nevertheless, specific binding of Vip3 proteins to lepidopteran BBMV has been shown and it is generally accepted that binding to specific receptors is the basis for the specificity of Vip3 proteins [25,31,32]. Here, we have tested the binding of ^125^I-Vip3Aa to BBMV from susceptible and resistant *M. separata* insects to see whether we could find differences that could explain resistance. The lack of qualitative or quantitative binding differences between susceptible and resistant insects is in line with previous results with other Vip3Aa-resistant strains from other insect species for which Vip3Aa binding differences were not found [33,34]. Slower activation of Vip3Aa by midgut juice of *H. armigera* larvae has been shown in Vip3Aa-resistant insects, though because of the small differences with the susceptible insects, it did not seem to be the main reason for resistance [33]. Additionally, Vip3Aa-resistant *H. virescens* larvae showed dramatically reduced levels of membrane-bound alkaline phosphatase, but its involvement in resistance could not be demonstrated [34]. Therefore, unlike for Cry proteins, altered binding to membrane receptors seems not to be the main mechanism of resistance to Vip3Aa proteins, and other mechanisms should be explored [35]. Other steps in the mode of action of Vip3Aa, either prior (such as protease activation or peritrophic matrix sequestration) or after (pore forming, signal transduction, apoptosis, mitochondria disruption, etc.) binding to the membrane should be responsible for blocking the toxic action of the protein.

In conclusion, the biochemical basis of resistance to Vip3A proteins is still unknown and deserves further study. Additionally, alleles for Vip3Aa resistance seem to be relatively common, and thus, the use of Vip3A proteins alone, without combining them with Cry proteins, is not an appropriate strategy for the long-term implementation of this technology in pest control. The reported synergistic action of some combinations of Vip3Aa and Cry1 proteins [36,37] also favors the combined use of these two types of insecticidal proteins for better and most long-term use of the Bt-crops technology.

## 4. Materials and Methods

### 4.1. Insect Colonies

The laboratory colony of *M. separata* was established with 2500 eggs collected by putting an egg-laying substrate in a cornfield in Gongzhuling (Jilin Province, China) in June 2016. Insects were reared on an artificial diet at 24 ± 1 °C, photoperiod of 14:10 h (L:D), 70–80% RH. After two generations at the insectary (on an artificial diet), abundant eggs hatched (about 3000–5000 larvae) almost on the same day, and neonates were used to start bioassays and selection.

### 4.2. Source of Toxins

For bioassays and selection, the Vip3Aa19 protein was used, and it was provided by the Beijing DBN Biotech Center (DBNBC, Beijing, China). Trypsin-activated Cry1Ab and Cry1F (98% pure proteins) were produced and shipped by Marianne P. Carey, Case Western Reserve University, USA. The Vip3Aa protein used in the binding assays was Vip3Aa16 (NCBI Accession No. AAW65132) and the *Escherichia coli* clone containing the *vip3Aa16* gene was kindly provided by the Laboratory of Biopesticides, Centre of Biotechnology of Sfax (Sfax, Tunisia).

### 4.3. Bioassays and Selection Process

Bioassays were performed with an artificial diet [36] by the surface contamination method and conducted in 24-well (1.9 cm^2^) trays that contained approximately 1 mL of diet. Once solidified (about 10–15 min at room temperature), the diet was overlaid with an aqueous solution of Vip3Aa or Cry1 proteins dissolved in phosphate buffer saline (PBS) and allowed to air dry again. About six to nine different concentrations (48 larvae per concentration at generation 1 to 5, and 24 larvae per concentration from generation 6) of the tested proteins were used, which were chosen to produce between 20–90% mortality. One neonate larva (hatched during the previous 12 h) was added per well, covered with a perforated plastic membrane. Trays were maintained at the insectaria for seven days. The bioassay negative control consisted of just PBS; only bioassays in which the mortality in the control insects was less than 16% were considered. The mortality was recorded and judged dead if the larva did not move after repeatedly poked by a brush. Each bioassay was biologically duplicated twice on different dates.

Before starting the selection process, the toxicities of Vip3Aa, Cry1Ab, and Cry1F on *M. separata* (susceptible) were estimated (Appendix A). The selection pressure was thus adjusted to 60 µg/g for Vip3Aa19 (LC_95–99_), 60 µg/g for Cry1Ab (LC_75–80_) and 100 µg/g for Cry1F (LC_65–72_), and larvae (3000–5000 per generation and selection line) were maintained for 7–10 days. Larvae that survived and had at least molted to the second instar were gently picked out and transferred to a fresh diet without insecticidal protein until pupation. For each generation, neonates were monitored for susceptibility and subjected to selection, as described above. A sample of the population was maintained without selection to serve as a control, and the insects were reared in the same conditions as those subjected to selection.

### 4.4. Protein Purification for Binding Analysis

Conditions for bacterial culture and expression of the Vip3Aa16 were described previously [24]. The Vip3Aa16 from the supernatant of an *E. coli* cell lysate was purified using a HisTrap FF affinity purification column (GE Healthcare, UK), as described elsewhere [38]. Fractions (1 mL) were collected and those containing Vip3Aa were dialyzed against Tris-NaCl buffer (20 mM Tris, 150 mM NaCl, pH 9) overnight and then stored at −80 °C until used. The purity was checked by SDS–PAGE, and the concentration of protein was quantified by Bradford [39] before use.

### 4.5. Vip3Aa Radiolabeling

HisTrap FF purified Vip3Aa (25 µg) was labeled using 0.3 mCi of ^125^INa using the chloramine-T method, as previously described [24,40]. Then, the labeled protein was purified and separated from free ^125^I by a PD10 desalting column (GE Healthcare, UK). The purity of collected fractions was checked by subjecting an equal radioactivity signal (20,000 cpm) from each fraction to SDS-PAGE with further exposure of the dried gel (52 °C, 1 h) to an X-ray film at −20 °C. The purest ^125^I-Vip3Aa fraction was used for all the binding assays and stored at 4 °C. The estimated specific activity of the labeled protein was 3.9 mCi/mg.

### 4.6. BBMV Preparation

Fifth instar larvae of *M. separata* from both the susceptible (Ms-S) and the resistant (Ms-R) lines of the eighth and ninth selection generation (>3061-fold resistant) were dissected and the midguts (without the bolus content) in MET buffer (300 mM mannitol, 5 mM EGTA, 17 mM Tris, pH 7.5) were immediately frozen in liquid nitrogen and preserved at −80 °C until required. BBMV were prepared by the differential magnesium precipitation method [41], frozen in liquid nitrogen, and stored at −80 °C. The concentration of BBMV preparations was determined by Bradford [39] using bovine serum albumin (BSA) as a standard. The enrichment of apical membranes in the BBMV preparation was determined by measuring the activities of the apical membrane enzyme leucine aminopeptidase N, which provided an approximately fivefold enrichment.

### 4.7. Binding Assays with ^125^I-labeled Vip3Aa

Prior to being used in the binding assays, the Vip3Aa protoxin (both the ^125^I labeled and the unlabeled sample) was subjected to trypsin activation (trypsin from bovine pancreas, SIGMA T8003, Sigma-Aldrich, St. Louis, MO, USA) at 5% trypsin for 1 h at 37 °C. To determine the specific binding, the activated ^125^I-Vip3Aa (0.39 nM) was incubated for 1 h with different concentrations of *M. separata* BBMV (resuspended in binding buffer) at room temperature in a 100 uL final volume of binding buffer (20 mM Tris, 1 mM MnCl_2_, 0.1% BSA, pH 7.4). The reaction was stopped by centrifugation at 16,000× *g* for 10 min at 4 °C, and the pellet was washed twice with 500 µL of cold binding buffer (with centrifugation of 5 min after each wash). An excess of unlabeled Vip3Aa toxin (390 nM) was added to some samples to calculate the nonspecific binding. The radioactivity retained in the pellet was measured in a model 2480 WIZARD2 gamma counter (PerkinElmer, Downers Grove, IL, USA). The assay was repeated twice.

Homologous competition assays were performed as described above but using a fixed amount of BBMV (0.1 mg/mL) and increasing amounts of unlabeled activated Vip3Aa. The assay was repeated twice. The dissociation constant (*K*_d_) and the concentration of binding sites (*R*_t_) were calculated using the LIGAND program [42].

### 4.8. Statistical Analysis

Response (mortality)-dose data from bioassays were subjected to Probit analysis by the PoLoPlus V 1.0 program (LeOra Software Company, Petaluma, CA, USA) to generate the LC_50_ and LC_95_ values with 95% fiducial limits (FL), chi-square (χ^2^) and slope with standard errors (slope ± SE). LC_50_ values were considered significantly different if their 95% FL did not overlap. The level of resistance was expressed as the resistance ratio (RR_50_) and was calculated by the formula: RR_50_ = (LC_50_ of resistance line)/(LC_50_ of susceptible line).

## Figures and Tables

**Figure 1 toxins-13-00364-f001:**
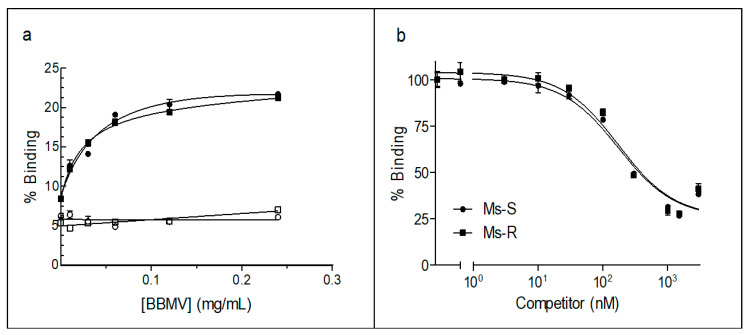
Binding of ^125^I-Vip3Aa (0.39 nM) to the susceptible Ms-S (circles) and the resistant Ms-R (squares) *M. separata* BBMV: (**a**) total (full symbols) and nonspecific binding (open symbols) at increasing concentrations of BBMV. The nonspecific binding was estimated in the presence of an excess of unlabeled Vip3Aa (390 nM); (**b**) binding of ^125^I-Vip3Aa at increasing concentrations of unlabeled Vip3Aa with 0.1 mg/mL BBMV.

**Table 1 toxins-13-00364-t001:** Response of *M. separata* to selection with Vip3Aa, Cry1Ab, and Cry1F.

Generation ^a^	Protein	LC_50_ (95% FL) µg/cm^2^	LC_95_ (95% FL) µg/cm^2^	RR_50_ ^b^	Slope ± SE	χ^2^
Ms-S	Vip3Aa	0.55 (0.26, 1.14)	213 (66, 850)	/ ^c^	0.71 ± 0.07	7.8
F9	Vip3Aa	>1684	− ^d^	>3061	−	−
Ms-S	Cry1Ab	0.14 (0.08, 0.23)	6.65 (3.04, 20.8)	/	0.99 ± 0.11	5.9
F8	Cry1Ab	0.39 (0.20, 0.60)	49 (15, 139)	2.7	0.76 ± 0.08	6.9
Ms-S	Cry1F	1.73 (0.70, 4.25)	271 (55, 8540)	/	0.75 ± 0.10	13.9
F9	Cry1F	2.60 (1.77, 3.69)	60 (35, 125)	1.5	1.29 ± 0.12	5.1

^a^ The values of the control population (Ms-S) are referred to as the last generation of selection. ^b^ RR_50_ = Ms-R LC_50_/Ms-S LC_50._ ^c^ / = Not applicable. ^d^ − = Impossible to obtain due to the high resistance level.

**Table 2 toxins-13-00364-t002:** Evaluation of cross resistance to Cry1 proteins in the Vip3Aa-selected population of *M. separata* (ninth generation).

Strains	Proteins	LC_50_ (95% FL) µg/cm^2^	RR_50_ ^a^	Slope ± SE	χ^2^
Ms-S	Cry1Ab	0.14 (0.08, 0.23)	/	0.99 ± 0.11	5.9
	Cry1F	1.95 (1.25, 3.08)	/	0.97 ± 0.11	7.8
Ms-R	Cry1Ab	0.39 (0.24, 0.62)	3	0.93 ± 0.12	4.1
	Cry1F	1.15 (0.93, 2.26)	/	1.15 ± 0.12	10.5

^a^ RR_50_ = Ms-R LC_50_/Ms-S LC_50._

**Table 3 toxins-13-00364-t003:** Equilibrium dissociation constant (*K*_d_) and concentration of binding sites (*R*_t_) of Vip3Aa with BBMV from susceptible and resistant *M. separata*.

Insects	*K*_d_ (nM)	*R*_t_ (pmol/mg)
Ms-S	40 ± 6	75 ± 22
Ms-R	41 ± 6	79 ± 17

## Data Availability

The data presented in this study are available in supplementary material.

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
