# Peer review of "The Rapid Evolution of Resistance to Vip3Aa Insecticidal Protein in *Mythimna separata* (Walker) Is Not Related to Altered Binding to Midgut Receptors"

_toxins, 2021, doi:10.3390/toxins13050364_

Round 1
Reviewer 1 Report
The paper presented is well structured.
The study contributes to understand the effect of distinct insecticidal proteins from Bacillus thurigiensis and highlights resistance mechanisms.
The conclusions concerning the use of VIP3A proteins alone as well as the recommendations of its combined use with Cry1 proteins are good tools to apply in Pest Forest Management and confers a practical approach to the present study.
I have some questions about the present work and some suggestions to its enrichment.
- Considering that resistance to VIP3Aa in Mythimna separata is not related to altered binding to midgut receptors:
- What could be the resistance mechanism?
- Is these absence of connection between the midgut receptors valuable only in the presents model (using separata) or is it possible to generalize in other insect species?
- Is there any difference in the synergy established between the protein pair Vip3Aa/Cry 1Ab and protein pair VIP3Aa/Cry1F?
- What is the basis for asserting that alleles for Vip3Aa are relatively common? It seems better to introduce there a bibliographic reference.
Author Response
Reviewer 1:
The paper presented is well structured.
The study contributes to understand the effect of distinct insecticidal proteins from Bacillus thurigiensis and highlights resistance mechanisms.
The conclusions concerning the use of VIP3A proteins alone as well as the recommendations of its combined use with Cry1 proteins are good tools to apply in Pest Forest Management and confers a practical approach to the present study.
I have some questions about the present work and some suggestions to its enrichment.
- Considering that resistance to VIP3Aa in Mythimna separatais not related to altered binding to midgut receptors:
- What could be the resistance mechanism?
We have added possible mechanism in lines 161 to 164.
- Is these absence of connection between the midgut receptors valuable only in the presents model (using separata) or is it possible to generalize in other insect species?
It is general. We already discussed this in lines 149 to 159.
- Is there any difference in the synergy established between the protein pair Vip3Aa/Cry 1Ab and protein pair VIP3Aa/Cry1F?
The synergies found previously (36) were around 2-fold in Vip3Aa/Cry1Ab and 6-fold in Vip3Aa/Cry1F, but there was no significant difference (by the overlap the 95% FL) between these two pairs.
- What is the basis for asserting that alleles for Vip3Aa are relatively common? It seems better to introduce there a bibliographic reference.
This is already explained in lines 75-79. But now, double checking it, you are correct that we did not make the appropriate estimate and we modified the value.
Reviewer 2 Report
The article “The rapid evolution of resistance to Vip3Aa insecticidal protein in Mythimna separata (Walker) is not related to altered binding to midgut receptors” describes the lab selection of resistant larvae from the oriental armyworm. This resistance develops after 9 generations against the Bt toxin Vip3A and unexpectedly not against other Cry toxins, i.e Cry1Ab and Cry1F. Because the resistance of an insect against a Bt toxin is due to a molecular mechanism in the midgut, they challenged the binding of Vip3A to the brush border membrane vesicles of the digestive tract of larvae. They conclude that the binding of the toxin is unaltered in the resistant strain.
Considering the length of this article, it should be considered more of a short communication than a regular article.
The authors should illustrate the Introduction with a picture of Mythimna separata (adult + larvae) as a single figure. See https://commons.wikimedia.org/wiki/File:Mythimna_separata_(catepiller).jpg and https://commons.wikimedia.org/wiki/File:Mythimna_separata_male.jpg
L65-66: “LC50 values of 0.86, 0.14 and 1.45 μg/cm2 for Vip3Aa, Cry1Ab and Cry1F, respectively”. Where have been done these LC50 values? In doi: 10.3390/toxins10110454, the LC50 appear in µg/g with 1.6, 6.4 and 14.4 for Vip3Aa, Cry1Ab and Cry1F, respectively. Can you justify the values used in the present study (add a supplementary figure or table to show how they were established).
Tables 1/2/3: the data are heterogenous. You should present in a single Table only the F0 and F9 generations for the three toxins. And in a separate supplementary Table F1 to F8 generations. This would clarify the reading of your data.
Figure 1
1a: please add the error bars which are not apparent on this figure. The statistical test is missing in the caption and in the mat & met section. It needs to be added. How many times did you replicate this assay ? This should be included in section 4.7 Binding Assays.
In the discussion, the only mechanism which could account for the resistance developed by an insect is the binding of Bt toxins to its receptor in the midgut. But authors should consider author mechanisms which have been described in resistant species: toxin solubilization, protoxin conversion into toxin, increased immune response through generations etc. this has not been discussed at all, since the authors have only considered the binding of the toxin to its receptor. This point should be more discussed.
Minor point
L331: separata
Author Response
Reviewer 2:
The article “The rapid evolution of resistance to Vip3Aa insecticidal protein in Mythimna separata (Walker) is not related to altered binding to midgut receptors” describes the lab selection of resistant larvae from the oriental armyworm. This resistance develops after 9 generations against the Bt toxin Vip3A and unexpectedly not against other Cry toxins, i.e Cry1Ab and Cry1F. Because the resistance of an insect against a Bt toxin is due to a molecular mechanism in the midgut, they challenged the binding of Vip3A to the brush border membrane vesicles of the digestive tract of larvae. They conclude that the binding of the toxin is unaltered in the resistant strain.
Considering the length of this article, it should be considered more of a short communication than a regular article.
We accept to convert it into a short communication.
The authors should illustrate the Introduction with a picture of Mythimna separata (adult + larvae) as a single figure. See https://commons.wikimedia.org/wiki/File:Mythimna_separata_(catepiller).jpg and https://commons.wikimedia.org/wiki/File:Mythimna_separata_male.jpg
We added the figure in Supplementary material
L65-66: “LC50 values of 0.86, 0.14 and 1.45 μg/cm2 for Vip3Aa, Cry1Ab and Cry1F, respectively”. Where have been done these LC50 values? In doi: 10.3390/toxins10110454, the LC50 appear in µg/g with 1.6, 6.4 and 14.4 for Vip3Aa, Cry1Ab and Cry1F, respectively. Can you justify the values used in the present study (add a supplementary figure or table to show how they were established).
We have added a table in Supplementary Material (Table S1) indicating these values
Tables 1/2/3: the data are heterogenous. You should present in a single Table only the F0 and F9 generations for the three toxins. And in a separate supplementary Table F1 to F8 generations. This would clarify the reading of your data.
Tables 1, 2 and 3 were combined as indicated. The original information has been moved to Suppl. Mat.
Figure 1
1a: please add the error bars which are not apparent on this figure. The statistical test is missing in the caption and in the mat & met section. It needs to be added. How many times did you replicate this assay ? This should be included in section 4.7 Binding Assays.
We added the error bars in the figure and the number of replicates performed. We did not perform any statistical test to compare the Kd and Rt values between colonies since the values were so similar, and not significantly different if we consider the standard error.
In the discussion, the only mechanism which could account for the resistance developed by an insect is the binding of Bt toxins to its receptor in the midgut. But authors should consider author mechanisms which have been described in resistant species: toxin solubilization, protoxin conversion into toxin, increased immune response through generations etc. this has not been discussed at all, since the authors have only considered the binding of the toxin to its receptor. This point should be more discussed.
We have added possible mechanism in lines 161 to 164.
Minor point
L331: separata
Done
Round 2
Reviewer 2 Report
Modifications have been brought to the draft as requested.
Minor
Figure S1. Mythimna separata
In Table 1 and Table 2: RR50 or RR50 (lower cas or font policy)